# Cardiac Remodeling and Repair: Recent Approaches, Advancements, and Future Perspective

**DOI:** 10.3390/ijms222313104

**Published:** 2021-12-03

**Authors:** Perwez Alam, Bryan D. Maliken, Shannon M. Jones, Malina J. Ivey, Zhichao Wu, Yigang Wang, Onur Kanisicak

**Affiliations:** 1Department of Pathology and Laboratory Medicine, College of Medicine, University of Cincinnati, Cincinnati, OH 45267, USA; alampz@ucmail.uc.edu (P.A.); jones2so@mail.uc.edu (S.M.J.); iveymj@ucmail.uc.edu (M.J.I.); wuzc@ucmail.uc.edu (Z.W.); wanyy@ucmail.uc.edu (Y.W.); 2Harrington Physician-Scientist Pathway, Department of Internal Medicine, University Hospitals Case Medical Center, Cleveland, OH 44106, USA; bryan.maliken@uhhospitals.org

**Keywords:** cardiac repair, regeneration, angiogenesis, fibrosis, MicroRNA, cell therapy

## Abstract

The limited ability of mammalian adult cardiomyocytes to proliferate following an injury to the heart, such as myocardial infarction, is a major factor that results in adverse fibrotic and myocardial remodeling that ultimately leads to heart failure. The continued high degree of heart failure-associated morbidity and lethality requires the special attention of researchers worldwide to develop efficient therapeutics for cardiac repair. Recently, various strategies and approaches have been developed and tested to extrinsically induce regeneration and restoration of the myocardium after cardiac injury have yielded encouraging results. Nevertheless, these interventions still lack adequate success to be used for clinical interventions. This review highlights and discusses both cell-based and cell-free therapeutic approaches as well as current advancements, major limitations, and future perspectives towards developing an efficient therapeutic method for cardiac repair.

## 1. Introduction

Cardiovascular diseases continue to be the leading cause of morbidity and mortality in the United States and worldwide [1]. Despite recent studies showing that neonatal murine and porcine hearts are able to regenerate via proliferating cardiomyocytes within the first few days after birth, adult mammalian cardiomyocytes (ACMs) lack meaningful endogenous proliferative potential. This lack of proliferation results in pathological repair mechanisms and fibrotic scarring after cardiac injury [2,3,4]. Various approaches have been tested to repair cardiac damage and restore heart function in the past few decades. These strategies include cell-based, non-cellular, induced adult cardiomyocyte proliferation and manipulation of cardiac remodeling. Though there has been significant success in delineating the mechanism of cardiac injury and protection against acute ischemic injury, an efficient therapeutic intervention is still unavailable. Therefore, it is of great importance to reanalyze the research development, interpret common/global outcomes, and identify the potential pitfalls in developing future therapeutic interventions. This review provides a comprehensive literature survey, encompassing multiple therapeutic approaches and deducing conclusions. Overall, this review highlights that the activation of neonatal cardiac pathways is a potential mechanism for cardioprotection in adult hearts and that the paracrine factors play a critical role in signal transduction for this reparative process.

## 2. Cell-Based Approaches to Cardiac Repair

Myocardial infarction (MI), defined as loss of functional cardiomyocytes, initiates a tumultuous pathway of fibrotic remodeling with scar formation resulting in decreased cardiac function and, ultimately, cardiac failure. All current therapeutic approaches aim to replace the injured cardiomyocytes or scar tissue with healthy and functional cardiac tissue to restore cardiac function. Since the heart is under constant mechanical load, immediate physiological stabilization of the heart is required after an injury to ameliorate functional deficiencies and preserve a maximal level of cardiac function. Therefore, upon injury, activating immune cells and fibroblasts play a critical role in clearing debris and protecting against myocardial stress by patching the area of damage with a fibrotic scar. Given knowledge of this physiological process, researchers logically proposed cell-based therapies as a means to replace lost cardiomyocytes and reverse fibrosis to reestablish cardiac function after injury. Various cell types with different cellular origins and functional characteristics have been tested in multiple preclinical and clinical trials including, skeletal muscle myoblasts (SMs) [5,6], bone marrow cells (BMCs) [7], embryonic stem cells (ESCs) [8,9], mesenchymal stem cells (MSCs) [10,11], and induced pluripotent stem cells (iPSCs) [12].

Cell-based cardiac repair studies began with SM transplantation in various animal models of MI [13,14]. SMs were a logical choice as one of the first cell type to be used since they are easy to obtain and readily differentiate into myotubes [5]. Initial SM transplantation preclinical studies showed improved cardiac function and reduced cardiac remodeling and scar size in animal models [6,15]. Likewise, SM transplantation was then observed to improve cardiac function in heart failure patients in several small non-randomized clinical trials [16,17]. Nevertheless, SM transplantation studies exhibited various limitations which restricted the use of SMs as a therapeutic intervention. The major limiting factor of SM transplantation therapy is the higher incidence of ventricular arrhythmias in the animals after cell transplantation, most likely due to the mismatch of electromechanical coupling between resident and transplanted SMs [18,19,20]. Additionally, SM transplantation in clinical trial studies with large randomized samples showed as neutral to modest cardiac protection with no improvement to left ventricular ejection fraction (LVEF) indicating that this protective effect was not translatable [21].

BMCs and ESCs also appeared to be promising alternatives to SMs for cardiac repair after MI with similar approaches. Like SMs, BMCs and ESCs cell transplants show improved cardiac function and decreased cardiac fibrosis in rodent models post-MI [22,23,24]. Similar to SMs, these cells are also easy to obtain and show the potential to differentiate into cardiomyocytes. However, these cells require pre-differentiation into mature cardiomyocytes prior to transplantation into an infarcted heart, as undifferentiated BMC or ESC carry the risk of developing teratomas after cell transplantation [25,26,27]. Unfortunately, to date, the lack of a standardized protocol to obtain a pure population of matured cardiac cells from BMCs or ESCs limits their therapeutic success [28]. Lastly, the risk of immunologic rejection because of genomic instability provides further concern for the utility of these cells [29].

More recent studies have brought attention to the use of MSCs and iPSCs that promise a greater cell fate plasticity and cardiomyocyte potential during cardiac repair post-MI, where cell transplants resulted in improved cardiac function in the injured heart in preclinical studies [30,31,32]. Of particular advantage in these studies was the ease of obtaining MSCs from various tissue types, albeit with varying efficacy [33]. Moreover, both MSC and iPSC have the potential of self-renewal and low immunogenicity, thus, are suitable for autologous transplantation [34,35,36]. Additionally, these cells showed an enhanced ability to differentiate into various cell types, including cardiomyocytes, and exhibit an ability to integrate into the host myocardium [37,38,39,40]. Despite being easy to obtain and propagate, complete differentiation of MSCs and iPSCs into cardiomyocytes continues to pose the most significant challenge of utilizing these cells therapeutically [41]. In fact, cardiomyocytes derived from MSCs and iPSCs show a variety of maturity and transcriptomic states similar to neonatal cardiomyocytes, which results in a heterogeneous population of undifferentiated cells [25,42]. Similar to BMCs and ESCs, MSCs and iPSCs also pose the risk of teratoma formation, limiting these cells’ therapeutic potential [43].

Alternative approaches such as using a heterogeneous mix of cardio-sphere-derived cells (c-kit+, CD105+, and CD90+) have been the basis for a randomized phase one clinical trial showing a modest improvement with reduced scar size, and increased viability of myocardium; however, in terms of cardiac function, neither end-diastolic volume, end-systolic volume, or LVEF showed major difference between groups by 6 months [44]. Another report identified a subpopulation of cardiac mesenchymal cells, categorized as c-kit expressing and slow adhering when cultured, that improved cardiac function with significant recovery of LVEF after MI [45]. Moreover, a repetitive administration of these cardiac progenitor cells showed a more significant cardioprotective effect and a restoration of cardiac function when compared to a single administration of an equal number of cumulative cells [46].

On the clinical front, the first multi-center, randomized, double-blind, placebo-controlled clinical trial named ‘Combination of Mesenchymal and c-kit+ Cardiac Stem Cells as Regenerative Therapy for Heart Failure’ (CONCERT-HF Trial) was recently completed. In this study, investigators administered a cell suspension including either or both c-kit+ cardiac stem cells and mesenchymal cells by direct transendocardial injection to heart failure patients. Briefly, patients receiving cell therapy showed mild but significant benefits against major adverse cardiac events and showed improved quality of life. More importantly, this study proved that the effects observed in cell-based interventions are due to either systemic or paracrine mechanisms [47]. In fact, various studies demonstrated that the improvement from these types of trials was not due to the de novo generation of cardiomyocytes, but a combination of paracrine effects and the host response to exogenous cell transplantations [48,49,50,51,52].

Moreover, the issues surrounding the stability and survivability of transplanted cells in a harsh ischemic and fibrotic myocardium are still the foremost limiting factor for the therapeutic success of cell-based approaches to repairing the heart. To this end, a number of studies have been performed with a primary focus to improve the intra-myocardial delivery and engraftment of cells. For example, a recent report successfully demonstrated enhanced engraftment of ESC-derived cardiomyocytes in the heart using an injectable nanomatrix gel [53]. Furthermore, improved survivability of ESC-derived cardiomyocytes upon encapsulation with nanomatrix gel resulted in reduced scar formation and retention of LVEF in treated animals compared to controls. Another study utilized platelet surface markers to label cardiac stem cells (CSC), enhancing their recruitment in the injured heart and improving retention of labeled CSC into the infarcted region. This resulted in a decreased infarct size and improved cardiac function [54]. Moreover, mediation of gene transfer to bone marrow-derived MSCs (BMMSCs) via chemically modified nanoparticle, molecularly organic-inorganic hybrid hollow mesoporous organosilica nanoparticle (HMON), led to improved survivability of these cells in the infarcted heart after transplantation, which improved cardiac function with a complete recovery of LVEF values, reduced cardiac fibrosis, and increased angiogenesis [55]. Although several small, non-randomized cell-based studies have been conducted with these approaches, yielding some satisfactory success, extensive randomized studies fail to replicate these effects. Thus, a search remains for the ultimate cell type or a sustained delivery method for complete cardiac repair and restoration of normal function post-MI.

While various cell-based cardiac repair studies have advanced to clinical trials despite only modest success, these studies are critical and sufficient to show that a cell-based therapeutic approach is safe in patients [56]. Moreover, in light of a vast body of work reestablishing the lack of retention of transplanted cells after myocardial injection, recent cell-based interventions, showing significant cardiac protection with reduced major adverse cardiac events, suggest the presence of indirect endocrine or paracrine mechanisms. In fact, a recent comprehensive study showed that even without any cell transplantation, simply activating the innate immune system via injecting a potent activator, such as zymosan [53], can incite regional cardioprotection and improvement in heart function as well as a reduction in myocardial fibrosis [50,51]. Thus, either paracrine-inducing or non-cellular simulation-based approaches are becoming attractive for treating cardiac injuries and preventing myocardial remodeling.

## 3. Cell-Free Approaches for Cardiac Repair

Regardless of the rapid death of implanted cells in the injured myocardium, most studies show improved cardiac function and, vascularization, as well as reduced infarct size, which leads to the hypothesis that paracrine factors may have an important role in cardiac repair. It has been observed that all the progenitor and stem cells used in cell-based therapeutics release secretory factors that have a paracrine effect on the neighboring cells. Secretome analysis revealed that these exosomes are concentrated alongside pro-angiogenic, pro-survival, proliferative, and immunogenic factors [57,58,59,60,61,62,63]. Interestingly, the content and composition of the secretome are highly dependent on the parental cell type and the physiological or pathophysiological condition. For example, one study showed that GATA-4 overexpressing MSCs secreted paracrine factors, which augmented angiogenesis and improved cell survivability in the ischemic myocardium [62]. In contrast, another study showed that overexpression of the pro-angiogenic miR-126 in MSCs leads to secretomes enriched with pro-angiogenic and pro-survival factors. Further mechanistic analysis of miR-126 over-expressing cells revealed increased expression of Notch ligand Delta-like (Dll)-4 is believed to play a key role in improving angiogenesis and blood flow in the infarcted heart, resulting in improved cell survivability and cardiac function with significantly higher LVEF values in miR-126 over-expressing cell treatment group compared to MSC alone [64]. Similar results were obtained in animals treated with secretomes isolated from human adipose-derived stem cells post-MI [61].

Recent reports demonstrate that reprogramming of stem and progenitor cells leads to a subsequent change in molecular composition and functional characteristics of secreted paracrine factors, which could be a plausible tool to be utilized as a therapeutic intervention for cardiac repair [59]. An in silico functional screening analysis of human secretomes demonstrated secretion of pro-angiogenic and pro-survival proteins in patients following acute MI. The study further showed endogenous expression of myeloid-derived growth factor (MYDGF) from bone marrow-derived monocytes and macrophages, which had a cardioprotective role after MI [59]. Moreover, treatment with cardiomyoblast-conditioned medium (H9c2) containing growth factor TNF-α with hypoxia preconditioning led to an enriched expression of anti-inflammatory proteins, angiogenic factors, and migratory cytokines in exosomes isolated from rat BMMSCs. Additionally, the secretomes isolated after treatment with TNF-α and hypoxia led to improved angiogenesis and cardiac function in rats post-MI [60]. However, prolonged hypoxic exposure of MSCs modulated energy metabolism and increased expression of monocarboxylate transporter-4 (MCT4) which can specifically increase cardiomyocyte death and thus decreased cardiac function with modest reduction in LVEF after MI [65]. 

As recent studies highlight, exosomes are on the rise as a therapeutic asset since they are a major constituent of the secretome and deliver most of its attributed protective features. These nano-sized vesicles are secreted by most cell types, including cardiomyocytes and iPSCs [66,67,68,69,70,71,72]. Exosomes were first studied for their role in the adaptive immune response [73]. Subsequently, they were observed to play a vital role in signal transduction, cell-to-cell communication, and other paracrine mechanisms [74,75,76]. Further scientific advancements utilized exosomes derived from different cell types to study their therapeutic significance in various disease conditions, including MI [77,78]. Exosomes have also been explored as a plausible tool to deliver small molecules, such as miRNA or therapeutic drugs, to areas of interest [79,80]. Given the limited success of developing efficient cell transplantation approaches, research focus was shifted to analyze the protective effects of paracrine factors secreted by engrafted cells into the infarcted heart. These studies revealed that most of the protective effects from cell-based therapies were indirect through secretomes or, more specifically, exosomes. Similar to secretomes, exosomes also exhibit the features of their parent cells, such as ESC-derived exosomes. Similar to ESCs, whose total miRNA content is comprised of about two-thirds of the miR-290 family, ESC-derived exosomes also express a high level of miR-290 [71]. Further investigation revealed that ESCs derived exosome-mediated delivery of miRNA-294 led to increased cell survival, angiogenesis, and proliferation, and thus improved cardiac repair and restoration of cardiac function post-MI [71].

Moreover, treatment with the exosomes isolated from cardiosphere-derived cells showed enhanced cardiac repair by improving cardiac function and decreasing infarct size in acute as well as chronic porcine MI models [81]. Furthermore, cardiac progenitor cells (CPCs) are known to regulate cardiac protection and repair. Notably, the exosomes derived from CPCs likewise increased cell survivability and proliferation of H9C2 cells through enhanced expression of Akt and activation of the Akt/mTOR pathway [82]. Interestingly, a pediatric study showed that exosomes derived from neonatal CPCs improved cardiac function and repair, whereas exosomes derived from CPCs of older children required hypoxia preconditioning to exhibit cardioprotective benefits. These effects included increased angiogenesis and reduced fibrosis, resulting in improved cardiac function in the infarcted heart [83]. Moreover, human pericardial fluid-derived exosomes carry let-7b-5p, which targets the TGFBR1 gene. An exosome-mediated delivery of let-7b-5p to endothelial cells led to improved angiogenesis which could have a protective effect in cardiac repair [84]. Furthermore, CD34+ HSC-derived exosomes express a high level of pro-angiogenic miRNAs, including miR-126 and miR-130, which improved vascular formation in the injured heart [85]. A miRNA sequence analysis between MSC and MSC-derived exosomes showed similar miRNA profiling, demonstrating a mechanistic similarity between MSC and MSC-derived exosome-mediated cardiac repair. This emphasized a greater therapeutic importance of MSC-derived exosomes over MSCs themselves for cardiac repair [86]. Other studies further demonstrated that exosomes derived from the hypoxia preconditioned MSCs show greater expression of various miRNAs involved in improved cell survival, angiogenesis, and reduced fibrosis, leading to enhanced cardiac repair in comparison to normoxic MSCs-derived exosomes [87,88,89,90]. Further analysis revealed that preconditioning CSCs with MSC-derived exosomes also results in improved cell survival and angiogenic potency of CSCs [91]. 

Regardless of the significant advancement and exciting results in cell-free therapy for cardiac repair, none of these studies were able to demonstrate significant cardiac regeneration even in small animals [49,92,93]. In addition to that, there are various other basic questions and limitations which need to be answered before improved therapeutic success with these methods. One of the major limitations is that the extracellular vesicles or exosomes require a direct intra-myocardial injection to the heart. Additionally, it is known that exosomes exhibit the features of the parental cell type, thus, it would be imperative to ensure that an autologous exosome does not carry any functional limitation when isolated from old cells.

## 4. Induced Cardiomyocyte Proliferation for Cardiac Regeneration and Repair

The senescent nature of adult mammalian cardiomyocytes (ACM) restricts their ability to proliferate and regenerate or repair after cardiac injury. However, zebrafish and neonatal mice demonstrate a propensity to regenerate cardiac tissue following injury, mostly facilitated by the proliferation of pre-existing cardiomyocytes [3,94]. Irrespective of the limited success of cell-based cardiac repair approaches, it remains of interest to induce the proliferation of endogenous ACM as a novel therapeutic approach to repair cardiac injury in adult animals. Importantly, in the last decade, proliferation studies with mammalian ACMs sufficiently demonstrated the scope of ACM proliferation through external interventions [95,96]. Reports demonstrate that the oxidative phosphorylation-based energy metabolism of ACMs leads to increased oxidative stress, resulting in elevated DNA damage response and, ultimately cell cycle arrest [97].

Contrary to the adult mammalian heart, a comparatively hypoxic environment of the fetal mammalian heart predominantly utilizes glycolysis for energy production and has the capability of cardiomyocyte proliferation [98,99,100]. Moreover, neonatal mouse cardiomyocytes exhibit fetal-like features during the first week of the proliferative window, whereas a hypoxic exposure elongates the proliferative window in postnatal mice [97]. Furthermore, one study observed that hypoxic exposure induced the proliferation of endogenous cardiomyocytes in adult mice, which improved left ventricular function, and enhanced cardiac repair after MI [101]. During the last decade, various genes and miRNAs have been identified and characterized for their roles in cardiac development as well as ACM proliferation. Genome-wide analysis revealed *Fam64a* as a novel regulator of the cell cycle in fetal cardiomyocytes under hypoxic conditions. However, the expression of *Fam64a* significantly decreased in the postnatal cardiomyocytes after its exposure to the oxygen-rich environment [102]. In addition, *ERBB2* plays an important regulatory role in cardiomyocyte proliferation during both embryonic and neonatal stages. The constitutive expression of *ERBB2* showed dedifferentiation, cardiomyocyte proliferation, and improved cardiac function after MI. Further mechanistic analyses demonstrated that *ERBB2* mediated dedifferentiation and proliferation through ERK and GSK3β/β-catenin signaling pathways [103]. TBx20 also plays an important role in cardiomyocyte proliferation and normal cardiac development through activation of multiple signaling pathways, which are also associated with ACM proliferation and cardiac repair after MI. These include PI3K/AKT pathways, which improve ACM proliferation and survival [104,105]; HIPPO/YAP pathways, which improve cardiac regeneration, contractility, cardiac function, and survivability [106,107,108]; and BMP/Smad pathways, which improve cardiac repair, and function [109]. TBx20 also inhibits the expression of cell cycle inhibitors such as p21 and Meis1 by direct binding. Thus, an over-expression of TBx20 in infarcted hearts improves cardiac repair by inducing ACM proliferation [110].

Genome-wide functional screening identified forty miRNAs with the potential to induce neonatal cardiomyocyte proliferation in mice as well as rats. Moreover, this screening identified miR-590 and miR-199a for their potential to induce cardiomyocyte cell cycle re-entry in both neonatal and adult mice or rats [111]. Similarly, miR-210 overexpression in adult cardiomyocytes promoted cell survival, proliferation of ACMs, reduced fibrosis, and increased angiogenesis [112].

Furthermore, a downstream target, miR-1825, was identified as a promising nodal regulatory candidate to induce ACM proliferation in mice and rats [113]. miR-1825 directly targets NDUFA10, a key gene involved in the mitochondrial electron transport chain, and over-expression of miR-1825 leads to decreased mitochondrial number and integrity. These findings agree with reports demonstrating that increased oxidative phosphorylation leads to elevated ROS production, DNA damage, and subsequently, cell cycle arrest [97,98]. Further analysis revealed that miR-1825 mediated the enhanced expression of miR-199a, which directly inhibited two cell cycle inhibitors, *Retinoblastoma1* (*Rb1*) and *Meis homeobox 2* (*Meis2*). Over-expression of miR-1825 inhibits the expression of *Rb1* and *Meis2,* which results in ACM cell cycle re-entry, increased angiogenesis, and cell survival [113]. In our most recent study, we demonstrated the feasibility of transiently inhibiting both of these cell cycle arrest regulators, which resulted in the proliferation of ACMs, improved survivability during hypoxia, and resulted in cardiomyocyte-specific paracrine signaling similar to the juvenile cardiomyocytes that promoted angiogenesis and prevented fibrosis, subsequently improving cardiac repair post-MI [114]. 

Another study demonstrated that forced expression of the cardiac hypertrophy-associated protein, Dyrk1a, results in hypo-phosphorylation of Rb1. This active form of Rb1 suppresses the expression of transcription factor E2f-dependent cell cycle-associated genes and thus, inhibits cell cycle progression [115]. Meis1 is another transcription factor, which has an inhibitory effect on cell cycle progression [116,117]. Overexpression of *Meis1* has a functional association with the senescent nature of postnatal cardiomyocytes [116], whereas direct translational inhibition of *Meis1* through miR-548c-3p, miR-509-3p, or miR-23b-3p is conducive to inducing ACM proliferation [118]. Moreover, Meis1 deletion led to an increased proliferative window in postnatal mice and improved cardiac repair and function in adult mice after MI [116]. Another study showed the direct targeting of miR-128 onto chromatin modifier *SUZ12*, which inhibited the expression of *p*^27^, and, thus, facilitated cell cycle progression. Additionally, cardiomyocyte-specific deletion of miR-128 showed improved cardiac regeneration and function through cardiomyocyte proliferation [119]. Recent scientific advancements utilized a cocktail of gene regulatory elements and sustained delivery tools such as slow-releasing hydrogels to develop an efficient therapeutic approaches [120]. Furthermore, a study using shRNA cocktail against *FoxM1, Id1*, and *Jnk3* genes to induce cardiac repair in adult animals, effectively inducing ACM proliferation in vitro as well as in vivo, resulted in improved cardiac repair, reduced fibrosis, and restored cardiac function after MI [121]. Another study used an injectable hyaluronic acid (HA) hydrogel for a sustained and localized delivery of miR-302 to the infarcted heart, showing a significant improvement in cardiomyocyte clonal expansion, improved left ventricular function, and, overall, improved cardiac repair [122]. Although there has been a substantial advancement in cardiomyocyte proliferation studies, a precise and accurate analysis of proliferating ACM is still a challenge. Current immuno-fluorescent-based proliferation analysis techniques are quite controversial, especially when all studies can show ACMs with a permanent cell cycle marker based on DNA synthesis phases, such as EdU or BrdU labeling after a presumed cell division; but they seldom can show a cardiomyocyte that is going through active cell division at cytokinesis with real-time imaging or other methods.

In fact, a recent study also showed the event of mitotic catastrophe, which happens due to DNA synthesis and mitosis without cytokinesis, falsely indicating ACM proliferation [123]. As most proliferative studies rely on DNA synthesis markers, it becomes hard to interpret if all cells passing through DNA synthesis are completing cytokinesis. Therefore, it becomes essential to generate and utilize new genetic cell division reporter mouse lines to accurately estimate proliferation events and lineage tracing in a real-time manner.

## 5. Heterogeneity among Endogenous Proliferative Potential of Cardiomyocytes

Adult cardiomyocytes are no longer considered a terminally differentiated post-mitotic cell type, as the studies described above found them to have a proliferative potential, albeit at a very low frequency [95,124]. Some studies bring about the idea of the existence of cardiomyocyte subpopulation with varying proliferative potential within the adult animal heart. These proliferative cardiomyocytes were detected in many organisms and shown to be in a mononuclear diploid state and maintain a lifelong capacity for proliferation [125,126]. Similarly, embryonic and neonatal cardiomyocytes in mammals remain in the mononuclear diploid cardiomyocyte state during the proliferative period [126]. However, when ACMs lose the ability to proliferate in adult animals, most of the ACMs acquire the multi-nucleated or polyploid state, likely due to mitotic catastrophe [123]. Still, a fraction of mono-nucleated diploid cardiomyocytes are present in the adult mammalian heart, which is believed to have the potential to proliferate and make new cardiomyocytes upon injury or during the course of life. A recent report demonstrated a variable frequency of the mono-nucleated diploid cardiomyocytes in different strains of mice, emphasizing the association of an animal’s genetic makeup with the frequency of these ACMs in the heart [127]. Moreover, this study further demonstrated a positive correlation between the frequencies of mono-nucleated diploid cardiomyocytes with the observed potential for cardiomyocyte proliferation. Additionally, association analysis revealed a cardiac-specific gene, Tnni3k, which is known to have an inhibitory effect on cell cycle progression at the mitotic phase and leads to polyploidy, now believed to be one of the factors controlling ACM proliferation [128].

Moreover, Tnni3k responds to oxidative stress, which could be a possible reason for cell cycle exit and polyploidy in ACMs [128,129,130]. As previously discussed, oxidative stress is one of the foremost factors influencing ACM cell cycle arrest. Since most ACMs remain in a polyploid state, the oxidative-stress-induced Tnni3k response could be of interest to study for its role in postnatal cell cycle arrest of ACMs. Additionally, this role suggests that proliferative cardiomyocytes in animals may have a distinct feature that can be used to differentiate them from non-proliferating ACMs based on their nuclear content and metabolic state. Further identification and characterization of these proliferating cardiomyocytes could help develop efficient and targeted regenerative and reparative therapeutics.

## 6. Conclusions

Cell-based therapies for MI were developed with the expectation to replenish the injured heart with an adequate number of functional cardiomyocytes in order to restore cardiac function. Initially, cell-based therapies showed exciting results in small animal models, but over time, various limitations were also identified. Despite the development of cell types with better survivability and sustained delivery techniques, most transplanted cells either die or wash away from the myocardium within a short time.

Moreover, none of the cell-based studies so far have demonstrated a substantial amount of endogenous cardiomyocyte proliferation that can lead to cardiac regeneration post-MI. Additionally, more recent studies hypothesize that most of the cardioprotective effects from cell-based therapies were via indirect endocrine or paracrine effects of transplanted cells in the heart. Stemming from this theory, the use of secretory factors, such as exosomes from various stem cell or progenitor cells, demonstrated a similar cardioprotective effect as observed through cell-based therapies.

Although exosomes have advantages over respective cell-based therapies, they exhibit the characteristics of their parent cells. Thus, no exosome has yet been characterized that can orchestrate clinically significant cardiac regeneration. 

Currently, induced proliferation of endogenous cardiomyocytes seems to be the most pragmatic approach to repair a cardiac injury. A significant amount of work has been done, and many more ongoing studies are focused on inducing endogenous cardiomyocyte proliferation by modulating gene expression in different disease models. However, the consensus is that paracrine signaling to improve the reparative cascade following cardiac events, either via cell-based or non-cell-based approaches have much better potential for translation. Moreover, a combinatory approach, including the manipulation of non-myocytes in the heart along with pro-regenerative methods, will pave the way to successful reparative cardiovascular therapies.

## Data Availability

Not applicable.

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
