# Peer review of "Cardiac Remodeling and Repair: Recent Approaches, Advancements, and Future Perspective"

_ijms, 2021, doi:10.3390/ijms222313104_

Round 1

Reviewer 1 Report

1. It would be helpful to have a table summarizing the various remodeling/repair strategies discussed throughout the review, providing a quick reference quick for readers. 

2. Missed opportunity to include tissue engineering approaches as possible repair strategies and platform for interrogating mechanistic inquiries. 

Author Response

  1. It would be helpful to have a table summarizing the various remodeling/repair strategies discussed throughout the review, providing a quick reference quick for readers. 

We have included a new table figure summarizing all the studies described in the manuscript.

  1. Missed opportunity to include tissue engineering approaches as possible repair strategies and platform for interrogating mechanistic inquiries. 

We thank the reviewer for bringing a very important point to our attention. We agree that tissue engineering has introduced several groundbreaking new advancements that improve the maturation and stability of cell-based therapies however after thorough consideration with our authors we have would like to opt to prepare a separate review article given our need to delve into the vast amount of published work we have not considered prior to giving the justice this section requires.

Reviewer 2 Report

Well written study.

Conclusion is too long, please revise this section, it should be more condensed  and informative

Reference nr 82 in INVALID - however in text there is citation with this number. Please explain.

Authors should present also studies with evaluation of ejection fraction after utilization of regenerative medicine

Author Response

1- The conclusion is too long, please revise this section, it should be more condensed and informative

We have made substantial revisions and concise this section.

2- Reference nr 82 in INVALID - however in the text there is a citation with this number. Please explain.

We have fixed this issue. The proper citation has been readded.

3- Authors should present also studies with an evaluation of ejection fraction after utilization of regenerative medicine

We have included specific details for LVEF values changes for many of the studies relevant to our conclusions. Please see edits.